# Creep Characteristics of a Strongly Weathered Argillaceous Sandstone Sliding Zone and the Disaster Evolution Mechanism of the Huaipa Landslide, China

Jinyu Dong [1], Yawen Zhao [1,*] , Handong Liu [1], Jiancang Zhao [2], Zhimin Zhang [2], Qiuhui Chi [1] and Jihong Yang [1]

1   College of Geosciences and Engineering, North China University of Water Resources and Electric Power, Zhengzhou 450045, China; dongjinyu@ncwu.edu.cn (J.D.); liuhandong@ncwu.edu.cn (H.L.); chiqiuhui1204@163.com (Q.C.); yangjihong@ncwu.edu.cn (J.Y.)
2   Henan Water Conservancy Survey Company Limited, Zhengzhou 450008, China; cain202306@163.com (J.Z.); yanty2023@163.com (Z.Z.)
*   Correspondence: b2020081810@stu.ncwu.edu.cn

**Abstract:** The creep characteristics of sliding zones strongly influence slope deformation and long-term stability, as well as the occurrence of landslide catastrophes. In this paper, large-scale triaxial creep tests were performed on the strongly weathered argillaceous sandstone sliding zone of the Huaipa landslide in the Henan Province, China, to study its creep characteristics and long-term strength in natural and saturated states. Three-dimensional numerical simulations were conducted to analyze the deformation creep law and catastrophic evolution mechanism of the slope after excavation and rainfall. The results show that the sliding zone underwent appreciable creep deformation prior to failure, and that the progression of specimen damage with an increasing stress level followed decay creep → steady creep → accelerated creep. The stress level played a decisive role in the creep deformation, with higher stress levels resulting in higher instantaneous displacement, creep displacement, and longer times required to reach steady creep. The stress level also determined the specimen's creep stage. When the stress level was low, the adjustment of the specimen's internal structure was dominated by air space compression and particle movement, whereas particle fragmentation mostly occurred at high stress levels. The long-term rock strength was approximately 62–66% of the instantaneous strength, the internal friction angle decreased by approximately 8° relative to the instantaneous strength, and the cohesion decreased by approximately 30%. The slope foot unloaded and deformed owing to the excavation of a bauxite mine at its front edge, after which the slope deformed via creep. The landslide disaster occurred when the deformation was significantly accelerated and the slope started to slide as a whole once the sliding zone became water saturated owing to continuous rainfall. The simulation results indicate that the landslide can be divided into a front edge bulging zone, central sliding zone, and trailing edge tension zone, which provides valuable insight on the creep deformation evolution process and the disaster mechanism of the landslide under the action of front edge excavation and rainfall.

**Keywords:** strongly weathered argillaceous sandstone; creep characteristics; particle breakage rate; long-term strength; disaster evolution mechanism

## 1. Introduction

Creep is one of the main causes of slope deformation and damage, and creep characteristics describe how the deformation and strength of rock and soil change over time. Slope deformation has a complex nonlinear relationship with time. Significant time-dependent creep properties have been reported in several landslides, including the Zentoku landslide in Japan [1], the creep landslide near Saga City, Kyushu Island, Japan [2], the creep landslide in the Gündoğdu district of Babadağ town in Denizli, Turkey [3], and a large number of dam slopes in the Three Gorges reservoir area in China [4,5]. Field investigations and

monitoring analyses of these landslides have revealed that slope excavation, rainfall, and the rise and fall of the water level in a reservoir are all important factors that affect creep characteristics [6].

In recent years, a series of technologies have emerged that enable people to monitor landslide motion patterns with temporal and spatial resolution [7,8]. However, the active mechanism of landslides can only be understood if the mechanical properties of the constituent materials of landslides are fully investigated [9]. Numerous studies have performed laboratory creep tests as the primary method to analyze the creep characteristics of rock and soil. Such tests generally include uniaxial compression creep tests [10–13], triaxial creep tests [14–18], and direct shear creep tests [19–21]. Rainfall alters the water content of rock and soil in slopes, which modifies their physical and mechanical parameters, creep characteristics, and affects. Triaxial creep tests are often used to study the effect of water content on the creep properties of rock and soil owing to their long testing times and better sealing ability than other tests. Liu et al. proposed a nonlinear creep damage model that considered the pore water pressure and used triaxial creep tests on saturated and dry sandstones to verify their model's validity [22]. Wang et al. investigated the effect of dry and wet cycles on the creep behavior of sandstone using triaxial creep tests [23]. Yang et al. conducted triaxial creep tests and analyzed the effects of freeze–thaw cycles, chemical erosion, and other factors on the creep parameters and long-term strength of rock [24,25]. Zheng et al. and Long et al. investigated the effect of water content and the drainage method on the creep properties of clay using triaxial creep tests [16,26].

Long-term strength and creep characteristics have also been a research focus of slope creep stability analyses [5,14,19]. The long-term strength of rock and soil refers to gradual (generally decreasing) changes in the material strength with increasing loading time. Numerous in-depth studies have been conducted on the long-term strength of landslides [27–29] and a consensus has been reached that a necessary condition for creep damage is when the shear stress of the rock and soil exceeds its long-term strength. Numerous tests have confirmed that the long-term strength of rock and soil directly undergoing creep damage is less than the peak strength.

Many studies have addressed the creep properties of landslide sliding zones [18,19,30], with most laboratory studies generally conducted on gravel-free soils [31–33] (with particle diameters finer than 2 mm, e.g., clay, silty clay). However, natural sliding zones often contain large particles with diameters greater than 2 mm, which cannot be ignored. Sun et al. and Wen et al. conducted creep tests on sliding zone specimens containing gravel and showed that the proportion of large particles significantly affected the creep behavior of the sliding zone [34,35]. Nevertheless, the use of large particles in creep behavior studies on sliding zones is scarcely reported.

Existing research on creep in sliding zones has focused on homogeneous samples or samples with small grains owing to size limitations of the test instrument. However, the creep characteristics of sliding zones are clearly affected by both the sample size and particle size. In the Huaipa landslide, strongly weathered rock formed a secondary weak layer with low mechanical strength and poor seepage properties. Coupled with the time effect, the strength of strongly weathered rock can be continuously compromised and develop into a sliding zone, thus threatening the slope safety. The deformation characteristics and disaster mechanisms of sliding zones with strongly weathered rock containing gravel particles are still unclear. To further investigate this question, large-scale triaxial creep tests and three-dimensional numerical simulation analyses were performed to study the creep characteristics of the sliding zone of a strongly weathered argillaceous sandstone and the mechanism of disaster evolution of a landslide under excavation and rainfall, taking the Huaipa landslide in Henan Province, China, as an example.

## 2. Huaipa Landslide Profile

Located in the western part of Henan Province, the Huaipa landslide is adjacent to the Yellow River in the north. The landslide within a low mountainous area of the Banyan

Mountain. The terrain is high in the north and low in the south. The Yellow River winds from west to east, with both sides of the river banked by high mountainous slopes in a steep, narrow valley. The line from the Heihu Mountain to the Koumen Mountain is the natural drainage divide between the Yellow River and the Luo River, and the more open Mianchi Basin is to the southeast. To the south and east of the landslide is a circular mountain, with steep terrain on its back edge, the 30–40 m deep Xipo ditch on its west side, and the Yellow River on its north side. The original slope topography had a slope degree of 11°–30°, and the landslide slope body had thick loose slope deposits. The lower slope was mostly excavated by mining; the mine pit is 220 m long from north to south and 180 m wide from east to west, with an average excavation depth of 20.0 m and an excavation volume of approximately 700,000 m³ (Figure 1a).

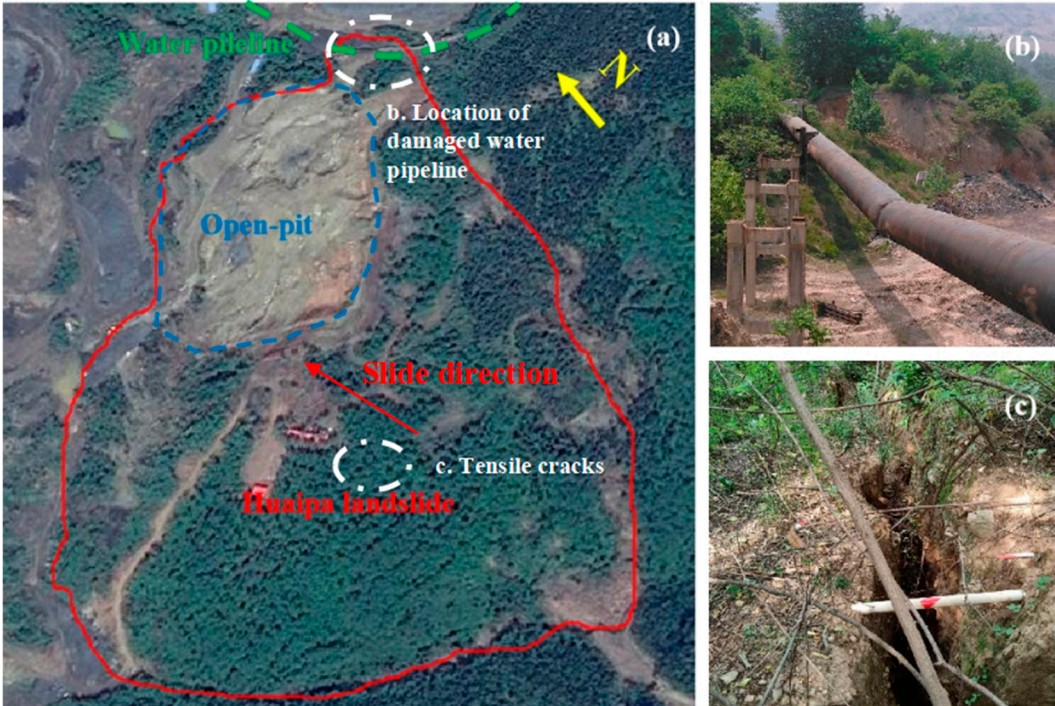

**Figure 1.** Geomorphology of the Huaipa landslide. (**a**) Topographic and geomorphological setting of the landslide area and slide direction. (**b**) Photographs of the damaged water pipeline at the foot of the slide. (**c**) Tensile cracks in the middle of the landslide.

The Huaipa landslide was in a creep–slip and deformation state for a long time as a result of the influences of mining and rainfall. At 6 a.m. on 2 May 2014, the slope was riddled with cracks and an overall slide occurred, causing serious damage to the 1# and 2# aqueducts of the Huaipa Yellow River water pumping project (Figure 1b,c). The aqueducts are the main water supply channel for industrial and drinking water for the urban residents in the counties of Yima and Mianchi.

### 2.1. Distribution and Deformation Characteristics of the Landslide

The landslide was controlled by the Meiyaogou fault, which is an F1 fault and fan-shaped (Figure 2). The landslide was approximately 495 m long in the north–south direction and approximately 370 m wide in the east–west direction on average, with an area of approximately 200,000 m², an average thickness of approximately 20.0 m, and a volume of approximately 4,000,000 m³, making it a large mid-level landslide with a main slip direction of approximately 324°. The terrain is steeper at the trailing edge of the landslide and gentler in the middle. To the southwest of the front edge of the landscape area is a primitive landform with steep terrain. In the northwest direction, there is an aluminum

mining area with flat terrain and a 10–25 m scarp along the east and south sides of the mining pit boundary with a slope degree of approximately 45°.

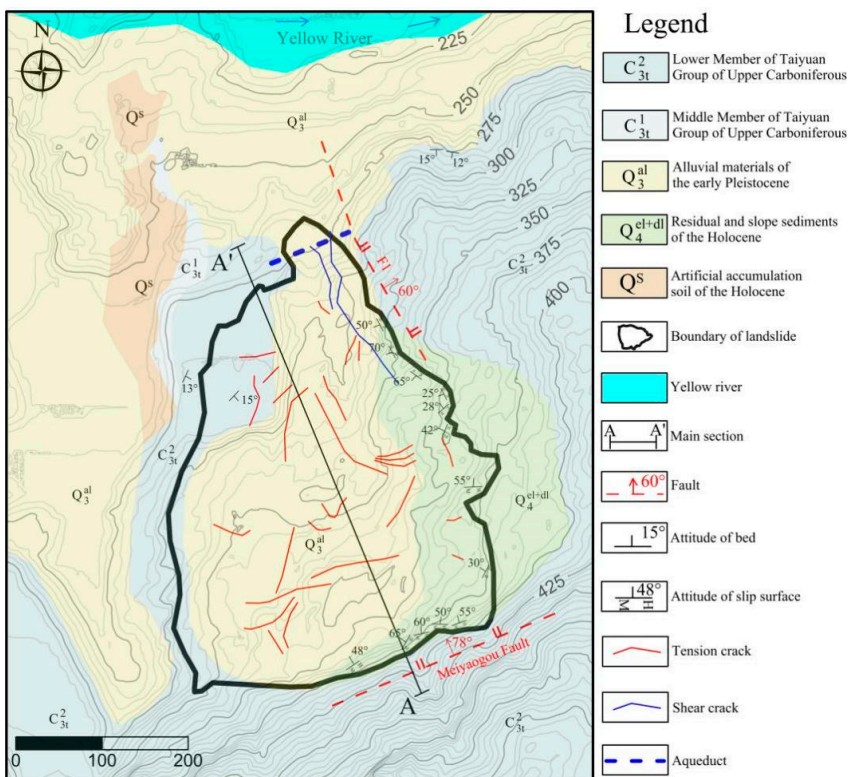

**Figure 2.** Engineering geological plan of the landslide area [36].

The results of a preliminary monitoring survey showed that the Huaipa landslide was related to bauxite mining on the front edge of the slope that took place in March 2013. The rainfall on the landslide area increased significantly in April 2014. The sliding mass was a loose mixture of soil and rock. Because of the small permeability coefficient of the bedrock, rainfall infiltrated along the sliding mass and gathered in the sliding zone area, causing it to become saturated. On 21 April 2014, the trailing edge of the slope began to slip and cracks opened with widths of approximately 1–3 m. The slope began to show significant deformation, and on 23 April 2014, the trailing edge of the slope appeared to slip and crack. At 6:00 a.m. on 2 May 2014, the cracks changed significantly and there was a large area of sliding. By 5:00 p.m., the landslide was basically stable.

### 2.2. Landslide Material Composition

According to the field investigation and drilling exposure, the trailing edge of the landslide tension zone and main sliding zones is mainly distributed in Quaternary eluvium and deluvium deposits of gravel clay, strongly weathered sandstone and mudstone, and strongly weathered argillaceous sandstone. The sliding zone thickness is generally 1.0–1.5 m. The residual gravel clay is loose, the gravel soil has high water permeability, and the cohesive soil in the gravel soil is easily softened in water. In contrast, the lower weathered bedrock has poor water permeability. The groundwater mainly flows downward along the bedrock. Under groundwater action, the landslide softens and slides along the weathered bedrock. The sliding zone near the soil–rock contact zone is generally wet and saturated, containing a high water content, and is relatively soft. Extensive water seepage occurred when the sliding surface was exposed by drilling. There were also crumpled or micro-inclined bedding planes, mirrors, and scratches on the sliding surface, as shown in Figure 3.

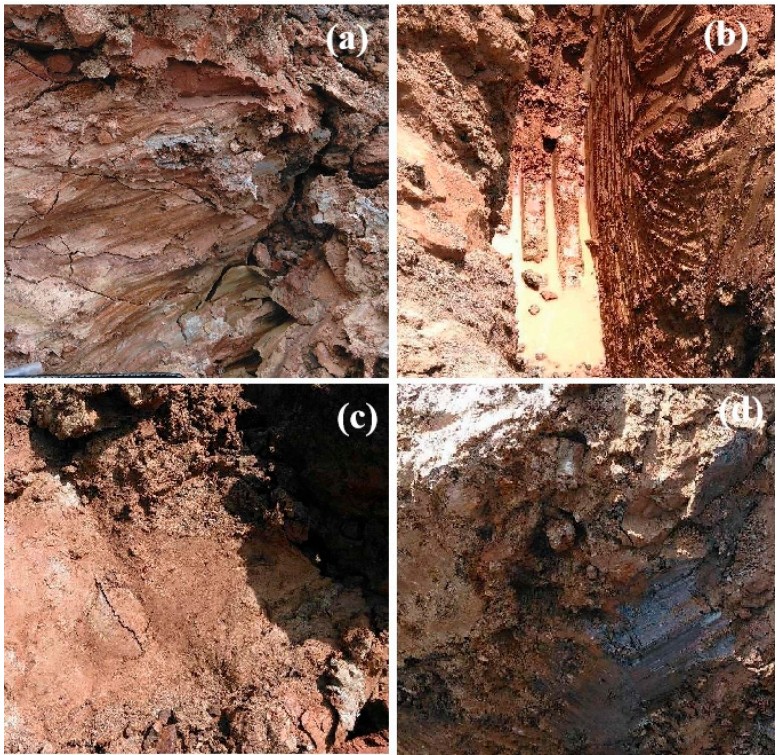

**Figure 3.** An inspection pit reveals the sliding zones. (**a**) Strongly weathered argillaceous sandstone in the sliding zone. (**b**) Water seepage of the sliding zones. (**c**) Scratches in the sliding zones. (**d**) Shallow shearing surface.

The sliding mass primarily consisted of residual gravel clay, strongly weathered sandstone, and mudstone; the sliding bed was primarily argillaceous sandstone; and the sliding zone was primarily distributed within strongly weathered argillaceous sandstone. The deformation and fracture properties indicate that the landslide can be divided into a bulging zone at the front edge, a fractured zone in the middle section, and a tension zone at the trailing edge (Figure 4). Table 1 lists the rock composition and thickness of each section of the landslide.

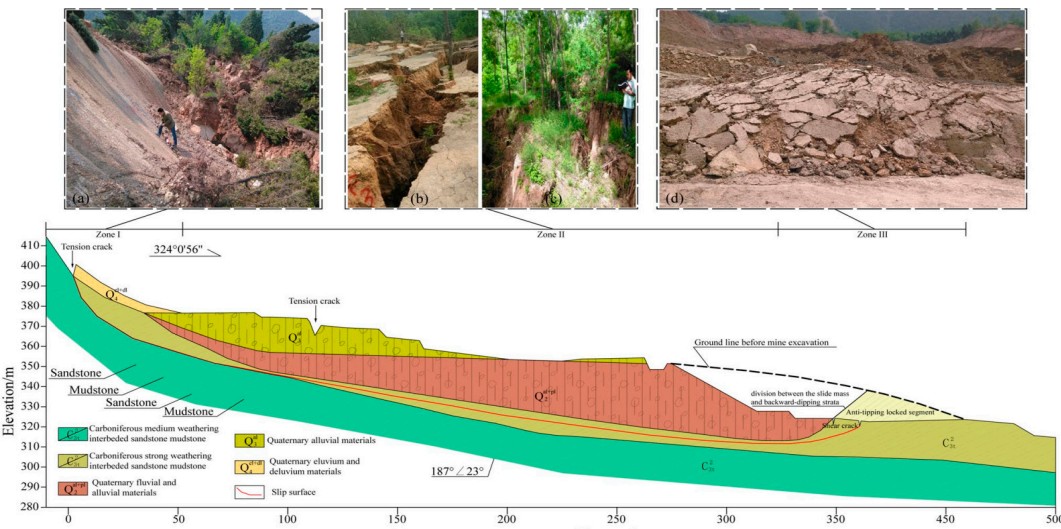

**Figure 4.** Engineering geological profile and photographs of the Huaipa landslide, showing the subzones of (**a**) the rear part of the slide and oblique trees, (**b**,**c**) tension cracks in the middle region, and (**d**) the distension zone in the front part of the slide [36].

**Table 1.** Structure of the Huaipa landslide.

| Subzone of Slide | Slide Mass | Slide Zone | Slide Bed |
|---|---|---|---|
| Front part | Quaternary eluvium and deluvium deposits, gravel clay, and strongly weathered sandstone and mudstone Thickness~10.0 m | Strongly weathered argillaceous sandstone Thickness~0.1–0.3 m | Anti-tipping mudstone and sandstone Dip angle = −19.8° to 1.2° |
| Middle part | Quaternary eluvium and deluvium deposits, gravel clay, and strongly weathered sandstone and mudstone Thickness~14.0–30.0 m | Strongly weathered argillaceous sandstone Thickness~1.0–1.5 m | Mudstone and sandstone Dip angle = 6.3°–14° |
| Rear part | Quaternary eluvium and deluvium deposits and gravel clay Thickness~13.0–29.0 m | Strongly weathered argillaceous sandstone Thickness~0.2–0.5 m | Mudstone and sandstone Dip angle = 17.8°–52° |

## 3. Creep Characteristics of the Strongly Weathered Argillaceous Sandstone

### 3.1. Large Triaxial Creep Test Scheme

The sliding zone of the Huaipa landslide was primarily strongly weathered argillaceous sandstone; some blocks were even breakable by hand. The fine-grained soil below 5 mm was reddish brown, mostly silt and clay, with a thickness of 1.0–1.5 m, and a natural moisture content of 13% (Figure 4). The sliding mass of the Huaipa landslide was eluvium and deluvium deposits of gravel clay with high permeability, whereas the bedrock had a low permeability. Under continuous rainfall, percolating rainwater tends to gather in the sliding zone and seepage flow occurs downward. Water permeation was visible upon exposing the sliding zone via drilling. The physical properties of the sliding zone are given in Table 2. The original grain size distribution curve of the specimen is shown in Figure 5.

**Table 2.** Physical property indicators of the strongly weathered argillaceous sandstone.

| Natural Moisture Content/% | Density/(g/cm³) | Specific Gravity | Liquid Limit/% | Plastic Limit/% | Plasticity Index |
|---|---|---|---|---|---|
| 13 | 2.09 | 2.68 | 24.6 | 16.6 | 8.0 |

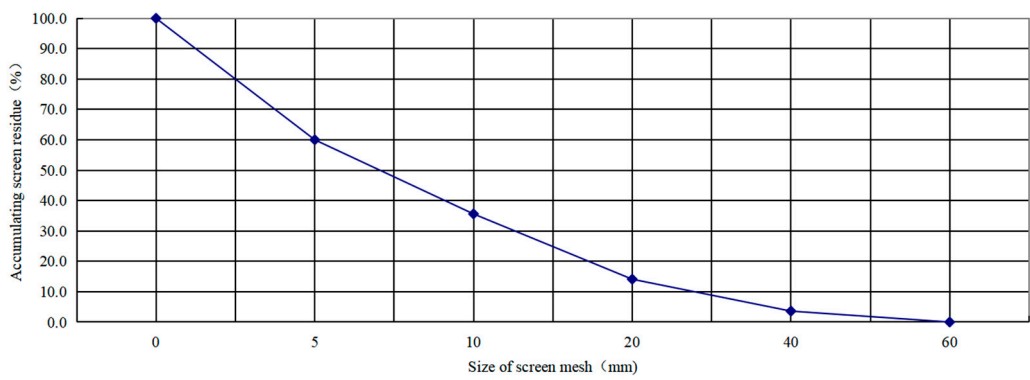

**Figure 5.** The original grain size distribution curve of the specimens.

To study the creep properties of the strongly weathered argillaceous sandstone and their influencing factors, creep tests were performed using a large triaxial creep apparatus under natural and saturated water content conditions at confining pressures of 100, 200, and 300 kPa (Figure 6). The creep specimens with 13% natural water content were classified as unconsolidated undrained shears (UU tests), and the saturated creep specimens were classified as consolidated drained shears (CD tests).

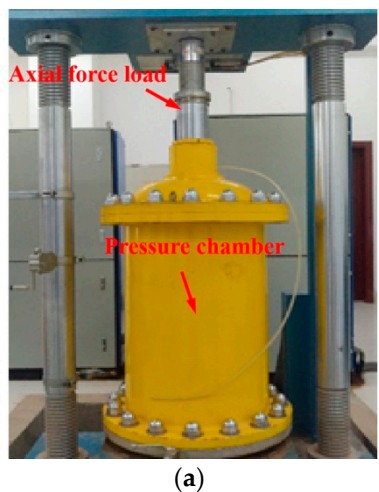
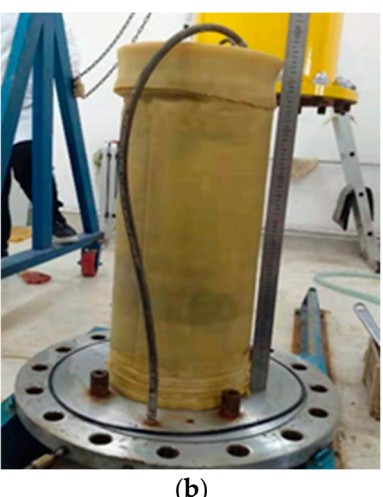

|          (a)          |          (b)          |

**Figure 6.** Large-scale SZLB-4 triaxial apparatus and a specimen of the strongly weathered argillaceous sandstone. (**a**) Triaxial apparatus, (**b**) Specimen.

Six sets of tests were performed on specimens with a diameter of 30 cm and height of 60 cm. Conventional triaxial tests were conducted to determine the ultimate deviator stress ($q_f$) at the time of damage. Graded loading was applied to reduce the creep test time and the dispersion of specimens. The selected creep stress levels were $0.2q_f$, $0.4q_f$, $0.6q_f$, $0.8q_f$, and $1.0q_f$, respectively. In the following, we abbreviate the stress level as 'SL' and the corresponding unused stress level is abbreviated as 'SL = 2'', 'SL = 4'', 'SL = 6'', and 'SL = 8''. A test was considered complete when the specimen axial strain reached 20%. The specimen deformation was then observed and the particle fragmentation was measured. Table 3 provides the specific creep loading scheme.

**Table 3.** Triaxial creep test scheme.

| Moisture Content | Confining Pressure/kPa | Deviator Stress/kPa |
|---|---|---|
|  | 100 | 125 → 250 → 375 → 500 → 620 |
| 13% | 200 | 180 → 360 → 540 →720 → 880 |
|  | 300 | 210 → 420 → 630 →840 → 1050 |
|  | 100 | 65 → 130 → 195 →260 → 320 |
| saturated state | 200 | 110 → 220 → 330 →440 → 530 |
|  | 300 | 150 → 300 → 450 →600 → 750 |

### 3.2. Analysis of the Experimental Results

#### 3.2.1. Creep Curve

Creep tests allowed the full creep process curve of the strongly weathered argillaceous sandstone to be obtained under different confining pressures, as shown in Figure 7. The creep curve was determined in four steps, and the specimen was damaged under the fifth load ($1.0q_f$). Accordingly, the creep curve was stepped once applying each shear load level under which the specimens show instantaneous displacement. Over time, the deformation speed gradually decreased and the curve tended to flatten.

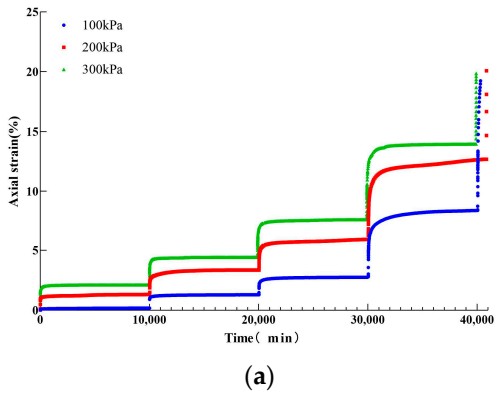
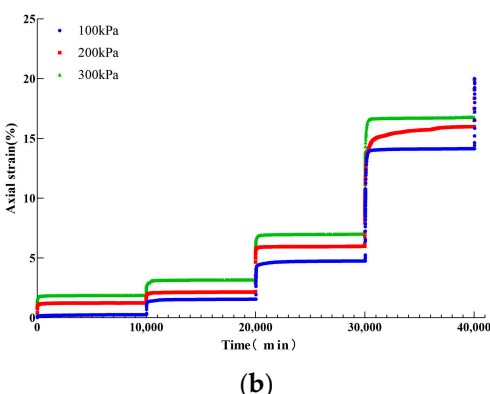

**Figure 7.** Graded loading curve. (**a**) Natural state, (**b**) Saturated state.

The full creep process curve (also called the graded loading curve) obtained from the test was converted into separate loading curves using Chen's loading method, as shown in Figure 8. Under each level of loading, the specimen first generated instantaneous strain, which then gradually flatted upon transitioning into creep strain. The instantaneous strain was mostly completed within 30 min and occupied the main part of the curve displacement. Although the displacement was relatively small, the creep strain represents the key part of the curve and can be used to determine the specimen's damage tendency. Research suggests that when the creep stress is less than the long-term strength, the soil sample exhibits a decaying creep mode; when the creep stress is greater than the long-term strength, the soil sample exhibits a creep or nondecaying creep mode as the strain rate increases [35,37].

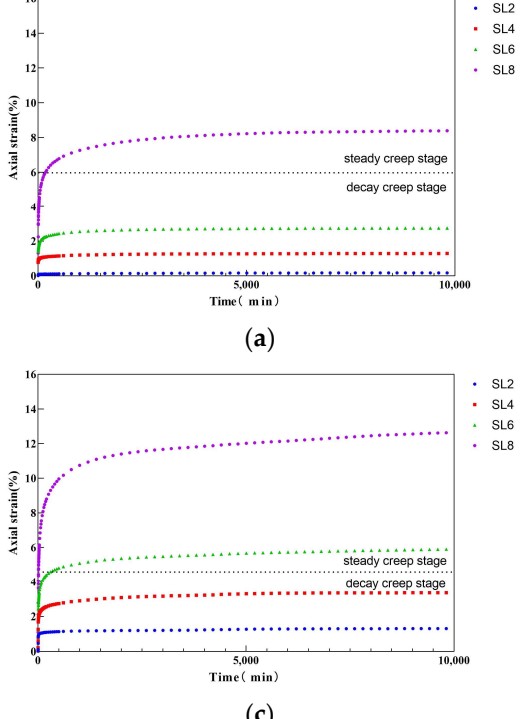
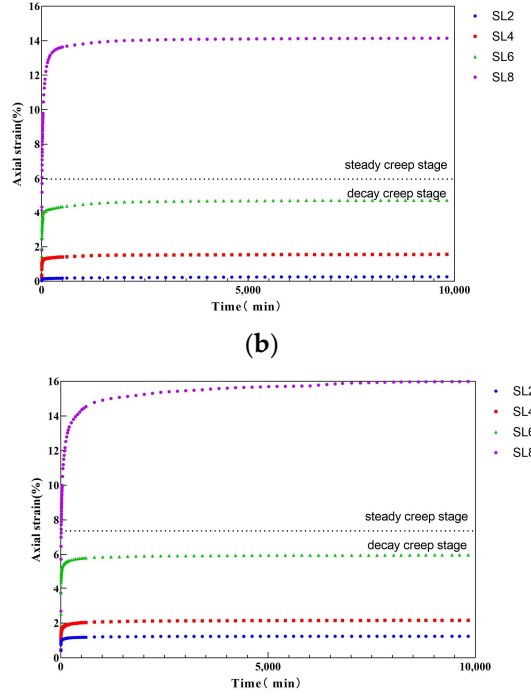

**Figure 8.** *Cont*.

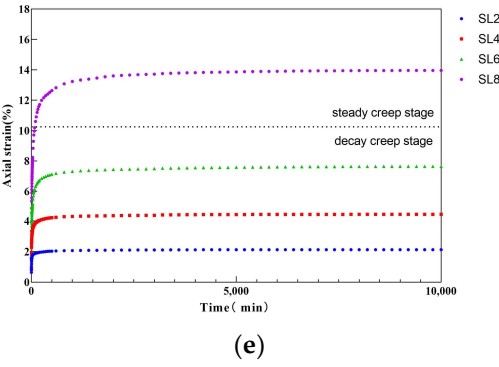

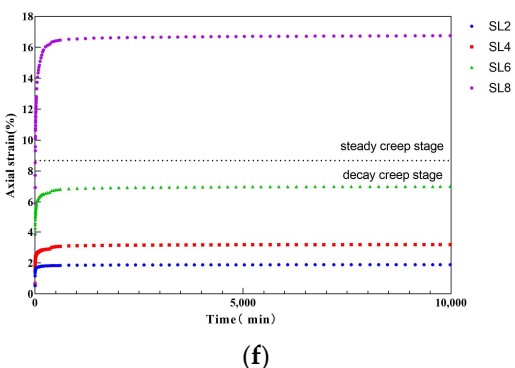

(**e**)

(**f**)

**Figure 8.** Loading curves. (**a**) Natural state with 100-kPa confining pressure, (**b**) Saturated state with 100-kPa confining pressure, (**c**) Natural state with 200-kPa confining pressure, (**d**) Saturated state with 200-kPa confining pressure, (**e**) Natural state with 300-kPa confining pressure, (**f**) Saturated state with 300-kPa confining pressure.

The full creep process can be divided into three stages (Figure 9): decay creep; steady creep; and accelerated creep [38]. The creep rate gradually decreased during the decay creep stage and remained approximately constant during the steady creep stage. Stable deformation occurred when the velocity was nearly constant. At lower stress levels, the specimens were dominated by instantaneous strain and stabilized at 3000–4000 min. The creep curve later approximated a horizontal steady state and the creep rate tended to zero. At this time, the specimen was in the decay creep stage and no damage occurred. As the stress level increased, the creep curve began to linearly increase at a lower creep rate and the specimen entered the steady creep stage. Upon maintaining this state, the specimen would have eventually been destroyed. The transition from decay creep to steady creep marked a turning point in the specimen's creep characteristics, and the corresponding deviator stress at this point was the specimen's long-term strength [39]. When a specimen was in the accelerated creep stage at higher stress levels, the strain rate increased rapidly and the sample rapidly sheared, causing damage.

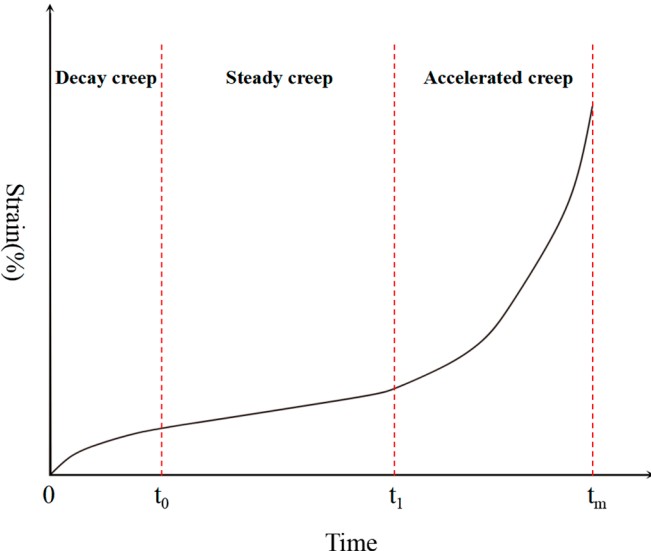

**Figure 9.** The full creep process curve.

### 3.2.2. Influencing Factors of the Creep Characteristics

Many factors affect the creep characteristics of rock and soil (e.g., mineral composition, particle gradation, moisture content, and stress level). The variables tested in the creep tests performed here included moisture content, stress level, and confining pressure.

(a) Moisture content Table 4 shows the ratios of instantaneous strain to total strain. The percentage of instantaneous strain to total strain is relatively small for creep specimens with 13% moisture content compared with the saturated specimens, ranging from 49% to 83%. The unsaturated specimens required longer times to reach stable deformation and showed a slow overall adjustment process. The instantaneous strain of the saturated specimens accounted for a relatively high percentage (56–89%) and the time required to reach stable deformation was short, exhibiting a relatively fast overall adjustment process. This is because water can reduce the cementation between particles, thus playing a lubricating role in the movement of soil and stone. The cementation effect and friction between particles in the saturated state are strongly weakened when an external force is applied, with a low resistance to movement, rotation, and overturning between particles, thus increasing the speed at which internal adjustments can be completed.

**Table 4.** Instantaneous strain and total strain.

| Time | Confining Pressure | Natural State | | | | Saturated State | | | |
|---|---|---|---|---|---|---|---|---|---|
| | | SL2 | SL4 | SL6 | SL8 | SL2 | SL4 | SL6 | SL8 |
| Instantaneous strain (30 min) Unit:% | 100 kPa | 0.10 | 0.99 | 1.90 | 4.57 | 0.15 | 1.23 | 3.76 | 9.79 |
| | 200 kPa | 1.00 | 2.15 | 3.38 | 6.15 | 1.03 | 1.69 | 5.24 | 9.99 |
| | 300 kPa | 1.76 | 3.40 | 5.23 | 8.24 | 1.67 | 2.54 | 5.87 | 12.81 |
| Total strain (10,000 min) Unit:% | 100 kPa | 0.17 | 1.29 | 2.75 | 8.40 | 0.26 | 1.56 | 4.75 | 14.15 |
| | 200 kPa | 1.31 | 3.34 | 5.91 | 12.64 | 1.24 | 2.15 | 5.98 | 15.99 |
| | 300 kPa | 2.12 | 4.43 | 7.65 | 13.96 | 1.89 | 3.18 | 7.01 | 16.75 |
| The ratio of instantaneous strain to total strain | 100 kPa | 56% | 77% | 69% | 54% | 56% | 79% | 79% | 69% |
| | 200 kPa | 76% | 64% | 57% | 49% | 83% | 78% | 88% | 62% |
| | 300 kPa | 83% | 77% | 68% | 59% | 89% | 80% | 84% | 76% |

Figure 7 shows that the creep specimens with a 13% moisture content exhibited better regularity than the saturated specimens at the same proportionally increasing stress level. The strain increase with stress level and deformation were similar, except at the fourth loading level, when those of the less-saturated sample were slightly larger. The strain increase at each stress level varied for the saturated creep specimens. At low stress levels, the displacement variation was small; at moderate stress levels, the displacement change tendency increased; and at high stress levels, the displacement increased rapidly. The shear strength of the saturated specimens was smaller than that of the natural specimens, and the displacement of the saturated specimens at high stress levels showed rapid growth and poor stability. Moisture content therefore has an important influence on the creep characteristics of strongly weathered argillaceous sandstone, and effective hydrophobic drainage is required for slope protection management.

(b) Stress level Stress level exerted the most notable effect on the specimen's creep characteristics. Higher stress levels were associated with larger instantaneous displacement, larger creep displacement, and longer times required to enter steady-state deformation. The stress level also determined the creep state of the specimens; when the stress level (SL) was low ($0.2 \leq SL \leq 0.4$) the specimens showed decay creep; at a medium stress level ($0.6 \leq SL \leq 0.8$) the specimens showed steady creep; and when the stress level was very high (SL = 1.0) the specimens exhibited accelerated creep and quickly experienced shear damage.

(c) Confining pressure The effect of confining pressure on the creep of the specimens was more complex. Higher confining pressures were associated with shorter times required for a specimen to enter stable deformation. However, at the same stress level, a higher confinement pressure resulted in larger creep deformation. This occurred because, with increasing confining pressure, the stress level corresponding to the deviator stress also increased, the stress state increased, the specimen deformation accelerated, and the overall deformation increased.

3.2.3. Particle Fragmentation Analysis of the Creep Specimens

Most contemporary studies concerning the strength characteristics of rock and soil are analyzed via macro-analyses, with few observations of the internal microscopic characteristics. However, studying the particle fragmentation in triaxial tests can help reveal the strength mechanism of soil–rock mixtures [40,41]. The particle fragmentation condition of the specimen was observed at the end of the tests completed in this study. Particle sieving analysis tests were performed to compare the changes in the particle gradation of the creep specimens before and after testing, and to analyze the influencing factors of the creep characteristics from the particle fragmentation characteristics. The particle fragmentation of each creep specimen is given in Table 5. The percentage of particle breakage ($B_g$) index proposed by Marsal (1967) was used to describe the particle fragmentation before and after testing [42]. The percentage of particle breakage was calculated as follows:

$$B_g = \Sigma \Delta W_k = \Sigma(W_{ki} - W_{kf}), \tag{1}$$

where $\Delta W_k$ is the difference between the content of each grain group before and after the test, $W_{ki}$ is the content of a certain grade of grain group before the test, and $W_{kf}$ is the content of the corresponding grain group of the same grade after testing. The grain size distribution curves of the specimens after the triaxial creep test are shown in Figure 9.

**Table 5.** Long-term strength and instantaneous strength.

| Moisture State | Strength | Confining Pressure | | | Strength Parameters | |
|---|---|---|---|---|---|---|
| | | 100 kPa | 200 kPa | 300 kPa | Cohesion | Angle of Internal Friction |
| 13% moisture content | Long-term strength | 405 kPa | 557 kPa | 662 kPa | 94 kPa | 23° |
| | Instantaneous strength | 617.5 kPa | 876.8 kPa | 1048.2 kPa | 130.9 kPa | 32.1 |
| | Proportionality | 66% | 64% | 63% | 72% | 72% |
| Saturated state | Long-term strength | 198 kPa | 347 kPa | 471 kPa | 21 kPa | 22.9° |
| | Instantaneous strength | 317.1 kPa | 522.7 kPa | 745.1 kPa | 30.5 kPa | 30.6° |
| | Proportionality | 62% | 66% | 63% | 69% | 78% |

We analyzed the effect of particle fragmentation on creep based on three aspects: moisture content; confining pressure; and stress level.

(a) Moisture content Figure 10 shows that the percentage of particle breakage in the saturated creep specimens was significantly lower than those in the natural state owing to the lubricating effect of water on the particle movement. Higher water content in the specimens was associated with more favorable movement, rotation, and overturning of the soil and stone. In the saturated state, water molecules effectively reduce the rigid contact between particles. The degree of particle fragmentation is significantly reduced when water molecules exist as a buffer between particles.

(b) Confining pressure Figure 10 shows that the percentage of particle breakage gradually increased with increasing confining pressure. When the moisture content was 13%, the percentage of particle breakage at 300 kPa was 1.7% higher than that at 100 kPa. Under saturated conditions, the crushing rate of the creep specimen at 300 kPa was 0.9% higher than that at 100 kPa. This is because increasing the confining pressure compresses the specimen and restricts the movement of soil and stone. The simultaneous increase of the contact surface between particles and contact force thus results in an overall increase in the degree of fragmentation.

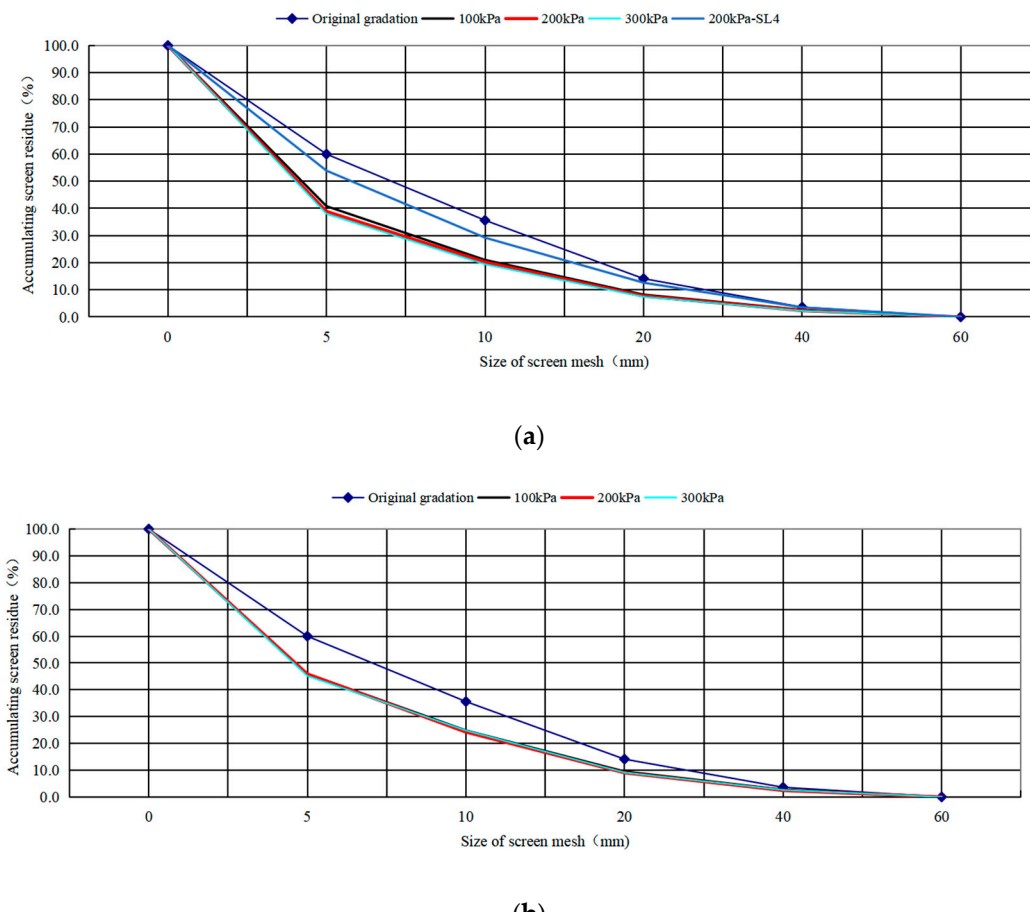

**(a)**

**(b)**

**Figure 10.** The grain size distribution of the specimens after the triaxial creep test. (**a**) Natural state (200 kPa-SL4 is the grain size distribution curve of the specimen after creep deformation had stabilized under a secondary load (0.4 qf) with 200 kPa), (**b**) Saturated state.

(c) Stress level To study the effect of stress level on particle fragmentation, the creep specimens with 13% moisture content were unloaded and sieved after creep deformation had stabilized under a secondary load ($0.4q_f$) and the percentage of particle breakage was calculated (Figure 10a). The results show that the particle fragmentation rate at low stress levels was not high; ~1/3 of the total fragmentation. This indicates that at low stress levels, adjustments of the internal specimen structure were dominated by void compression and particle movement, which rarely produce particle fragmentation. In contrast, particle fragmentation occurred mostly at high stress levels when there was more contact between the interior particles and the contact stress was high.

These results show that the essence of the creep process of the strongly weathered argillaceous sandstone involves the slow adjustment of internal particles and release of stress under the action of external forces. This is accompanied by complex strength loss and repair. Squeezing and fragmentation occur between the particles via sliding. The final state of particle adjustment occurs when the contact stress experienced by each particle under a certain stress level is minimized and equilibrates, movement or crushing ceases to occur, and rheological deformation tends to stabilize. The contact area and contact force between the particles are therefore key to particle fragmentation.

## 4. Long-Term Strength of the Strongly Weathered Argillaceous Sandstone

The main purpose of creep tests in engineering practice is to determine the long-term strength of rock and soil, and to analyze their stability under instantaneous stresses compared with the long-term strength. This provides a basis for evolutionary predictions.

In this paper, the long-term strength of the strongly weathered argillaceous sandstone was determined using the isochronous curve method based on the results of the triaxial creep tests.

### 4.1. Stress–Strain Isochronous Curves

The isochronous curve shows the stress–strain relationship curve at a certain moment under a certain confining pressure. Increasing the deviator stress caused the specimen to enter the steady creep stage; the deformation value of the specimen then increased sharply, causing the stress–strain isochronous curve to bend significantly. This bending point is assumed to correspond to the long-term strength. The isochronous curve at a confining pressure of 300 kPa in the saturated state is shown in Figure 11 as an example.

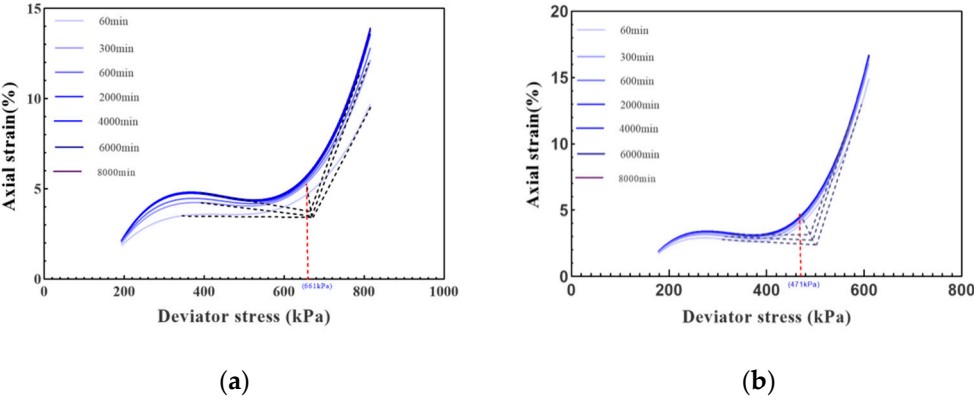

**Figure 11.** Stress–strain isochronous curves for a confining pressure of 300 kPa in the 13% moisture and saturated states. (**a**) 13% moisture content state, (**b**) Saturated state.

Figure 11 shows that the isochronous curves exhibit clearly nonlinear characteristics. The strains all show a change in pattern from flat to sharply increasing with increasing deviator stress. The curve increases more significantly for the saturation specimens than the samples with 13% water moisture. The stress–strain isochronous curve is also a double-inflection curve. The strain rate initially increases with increasing deviator stress, then stabilizes, followed by a subsequent increase. The first inflection point marks the transition from elastic deformation to viscoelastic deformation. The second inflection point is the transition from viscoelastic deformation to viscoplastic deformation. The deviator stress corresponding to the second inflection point is therefore taken as the long-term strength. The stress–strain isochronous curves also gradually increases with increasing deviator stress, which reflects the gradual strengthening of the time effect of the stress–strain in the specimen.

### 4.2. Long-Term Strength

Table 5 lists the long-term strength of each specimen determined by the isochronous curve method. The long-term strength of the saturated specimens is found to be much lower than that of the natural specimens with 13% moisture content. The long-term strength of both specimens increased with increasing confining pressure. A comparison of the instantaneous strengths in Table 4 indicates that the long-term strength of the specimen accounts for approximately 62–66% of the instantaneous strength. The internal friction angle of the creep specimen stabilized at 23–24°, a decrease of approximately 8° relative to that of the instantaneous strength. The cohesive strength of the natural and saturated specimens decreased by 36.9 and 9.5 kPa, respectively; a decrease of approximately 30%. The creep effect therefore has an important influence on the strength of the strongly weathered argillaceous sandstone. Accordingly, the long-term strength parameters should be fully considered in stability calculations for engineering projects, particularly those with high deformation requirements.

## 5. Three-Dimensional Numerical Simulations of the Huaipa Landslide

FLAC3D 6.0 software was configured with a creep constitutive model, and the Burgers–Mohr model was used to simulate the creep processes of landslides. The Burgers–Mohr model is a viscoplastic model with visco elasto-plastic deviatoric behavior and elasto-plastic volumetric behavior. Of these, the viscoelastic component corresponds to the Burgers model (containing a Kelvin substance in series with a Maxwell substance) and the plastic constitutive law corresponds to the Moore–Coulomb model. The parameters of the Burgers–Mohr model include the bulk modulus, cohesion, friction, Kelvin shear modulus, Maxwell shear modulus, Kelvin viscosity, and Maxwell viscosity. There is currently no complete theory that can accurately obtain these creep parameters from laboratory tests. Therefore, a calibration process is required to select the appropriate creep parameters. The sliding zone of the Huaipa landslide consists of strongly weathered argillaceous sandstone. A triaxial compression creep test simulation was first performed using FLAC3D and the creep parameters were calibrated with reference to the test results. A landslide creep simulation was then performed to analyze the long-term deformation characteristics and stability of the landslide.

### 5.1. Calibration with the Triaxial Compression Creep Tests

The thickness of the overburden layer on the sliding zone of the Huaipa landslide was approximately 15–20 m. The creep test curve with a confining pressure of 300 kPa was used to calibrate the creep parameters of the sliding zone. A series of triaxial creep tests were simulated to fit the Burgers–Mohr model creep parameters to the values listed in Table 6 using the trial difference method. The three-dimensional numerical model for the triaxial creep tests had a height of 60 cm and diameter of 30 cm, which is consistent with the test specimens. For the calculation, a confining pressure of 300 kPa was first applied to simulate the consolidation process prior to the creep test. After consolidation was completed, the corresponding deviator pressure was applied to simulate the triaxial creep test under different stress levels. The fitted curves are shown in Figure 12.

**Table 6.** Parameters determined using simulations of the triaxial creep tests.

| Moisture Condition | Bulk Modulus (kPa) | Cohesion (kPa) | Friction (°) | Kelvin Shear Modulus (kPa) | Kelvin Viscosity (kPa·min) | Maxwell Shear Modulus (kPa) | Maxwell Viscosity (kPa·min) |
|---|---|---|---|---|---|---|---|
| Natural | $1.38 \times 10^5$ | $0.094 \times 10^3$ | 23 | $8.44 \times 10^3$ | $7.2 \times 10^5$ | $0.24–0.7 \times 10^4$ | $1.728 \times 10^9$ |
| Saturated | $1.03 \times 10^5$ | $0.021 \times 10^3$ | 24 | $8.84 \times 10^3$ | $9.61 \times 10^4$ | $0.09–0.53 \times 10^4$ | $4.69 \times 10^8$ |

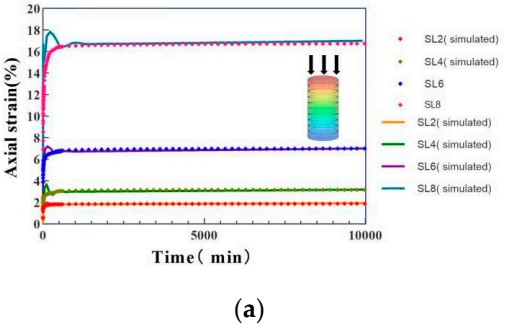 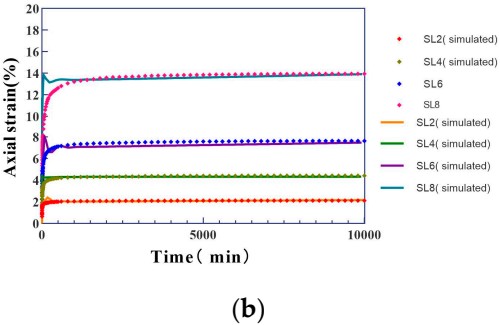

**Figure 12.** Fitting curves under a confining pressure of 300 kPa. (**a**) Saturated state, (**b**) Natural state.

### 5.2. Landslide Creep Deformation and Disaster Evolution Mechanism

To simulate the catastrophic creeping process of the Huaipa landslide under excavation and rainfall, a three-dimensional geological model of the landslide was established according to the field investigation and engineering geological survey, as shown in Figure 13. In

the calculation process, the elastic–plastic model was first used to analyze the deformation characteristics of the slope after the excavation of bauxite at the front edge of the slope in March 2013. The Burgers–Mohr model was used for the sliding zone to calculate the creep deformation of the slope under the condition of natural water content until April 2014. The creep deformation of the slope under the condition of sliding zone saturation caused by rainfall was then calculated until 2 May 2014. The calculated parameters are shown in Table 7.

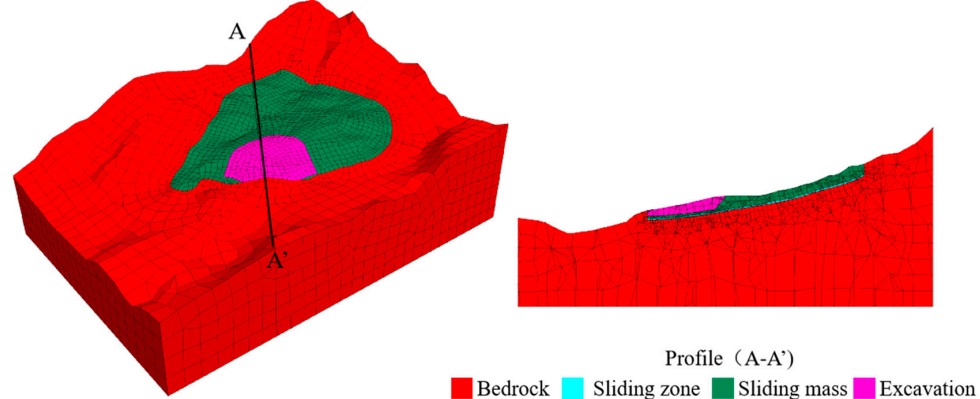

**Figure 13.** Calculation model (Profile A-A′ is the main sliding section).

**Table 7.** Mechanical parameters of the Huaipa landslide used for the numerical analysis.

| Parameters | Elastic Modulus (GPa) | Poisson Ratio | Unit Weight (kN/m³) | Cohesion (kPa) | Angle of Internal Friction (°) | Kelvin Shear Modulus (kPa) | Kelvin Viscosity (kPa·day) | Maxwell Shear Modulus (kPa) | Maxwell Viscosity (kPa·day) |
|---|---|---|---|---|---|---|---|---|---|
| Sliding mass | 0.25 | 0.32 | 21.2 | 150 | 32 | | | | |
| Bedrock | 75 | 0.26 | 27.1 | 310 | 39.7 | | | | |
| Sliding zone | 0.15 | 0.32 | 20.9 | 130 | 32.1 | | | | |
| Sliding zone | 0.15 | 0.32 | 20.9 | 94 | 23 | $8.44 \times 10^3$ | $5.00 \times 10^2$ | $6.50 \times 10^3$ | $1.20 \times 10^6$ |
| Sliding zone (saturated) | 0.105 | 0.33 | 21 | 21 | 23.9 | $8.84 \times 10^3$ | $6.67 \times 10$ | $1.00 \times 10^3$ | $3.26 \times 10^5$ |

Figure 14 shows the deformation evolution process of the slope from the natural state to the saturated state, as a result of the excavation of bauxite at the front edge and rainfall. The results demonstrate that the excavation of the bauxite mine at the leading edge of the slope in March 2013 destroyed the support at the slope foot. This caused unloading and produced large bulge deformation at the slope foot; however, the slope did not experience significant overall sliding at that time. Owing to the creep characteristics of the sliding zone in the natural state, the trailing edge of the slope was later damaged by tension and the slope foot sheared out. The deformation of the middle and rear of the slope increased significantly, with a maximum slope displacement of 15 cm over the 390 days prior to April 2014. Continuous rainfall in April 2014 caused the sliding zone to become saturated, and the slope deformation began to significantly accelerate. By 2 May 2014, the 422-day maximum displacement reached 21 cm. Finally, the slope began to slide as a whole and the large-area landslide occurred. According to the landslide body displacement deformation, the landslide can be divided into a front edge bulging zone (anti-slide section), a central sliding zone (main slide section), and a trailing edge tension zone. The simulation results are fully consistent with the field geological survey analysis shown in Figure 3.

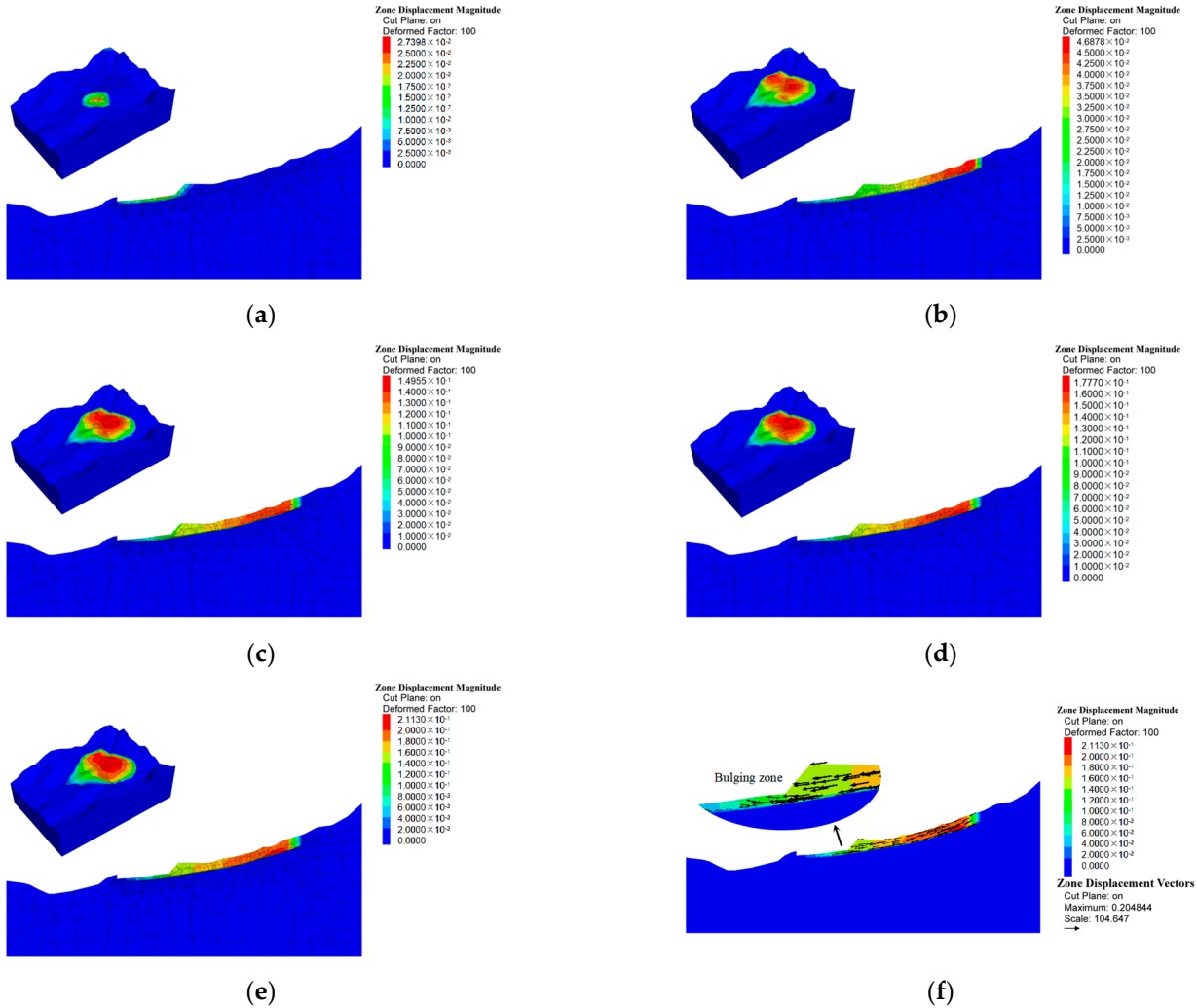

**Figure 14.** Deformation evolution process. (**a**) Excavation (March 2013) elastoplasticity analysis, (**b**) Natural state (creep time = 5 days), (**c**) Natural state (April 2014), (**d**) Saturation state (creep time = 395 days), (**e**) Saturation state (2 May 2014), (**f**) Displacement vector (creep time = 422 days).

Displacement monitoring points were set up along the landslide slope surface and location of the slip zone, as shown in Figure 15a. The monitoring results are shown in Figure 15b,c. Under natural conditions, the slope was found to be in a stable creep state and the maximum deformation rate of the slope was approximately 9 cm/year. After rainfall saturation, the landslide deformation increased rapidly. However, the landslide deformation rate decreased gradually over time and the landslide activity tended to stabilize.

The deformation curves and displacement of the slope surface and sliding zone are shown in Figure 16, indicating similar characteristics at both locations. The deformation of the slope body was largest at positions a2 and b2, and the deformation gradually decreased toward the slope foot. After excavation, the slope foot deformation was considerably reduced at positions a6 and b6, which indicates that the landslide was caused by the thrust load in the creep sliding process.

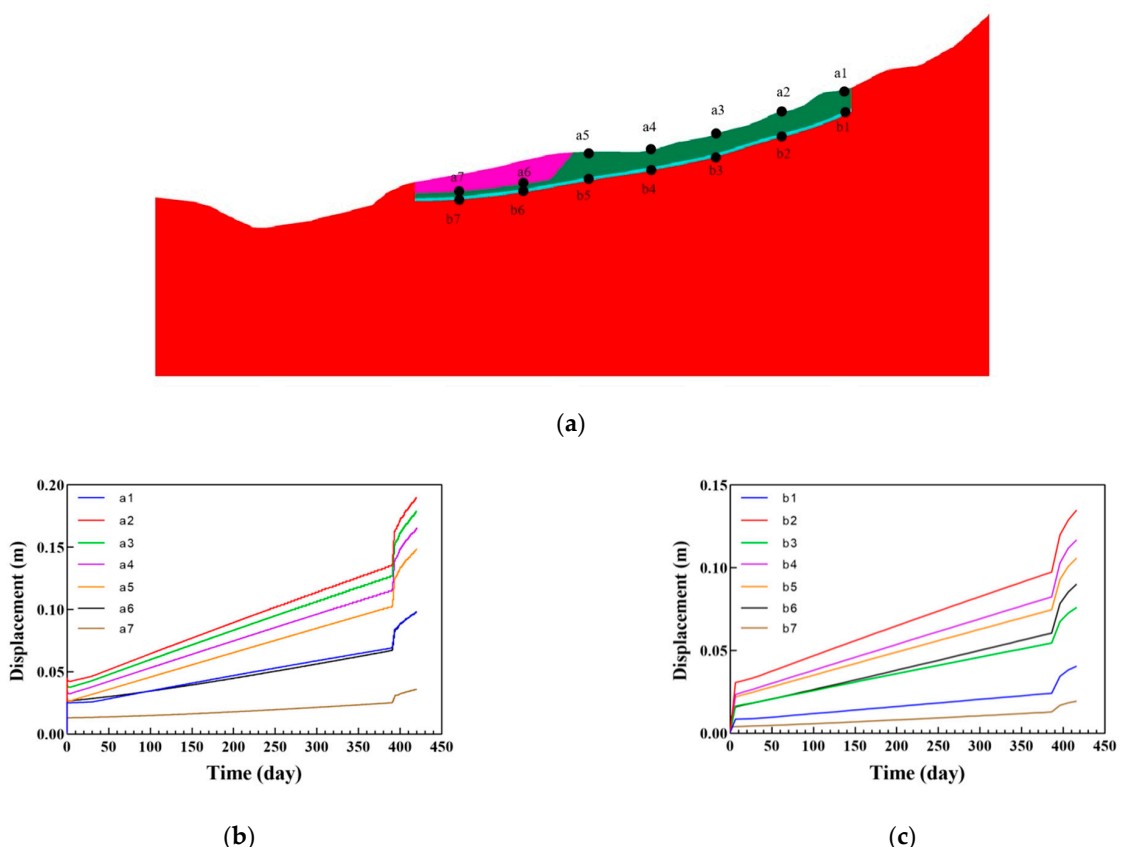

**Figure 15.** Monitoring curves of the slope displacement with time. (**a**) Monitoring point locations, (**b**) Slope surface displacement, (**c**) Sliding zone displacement.

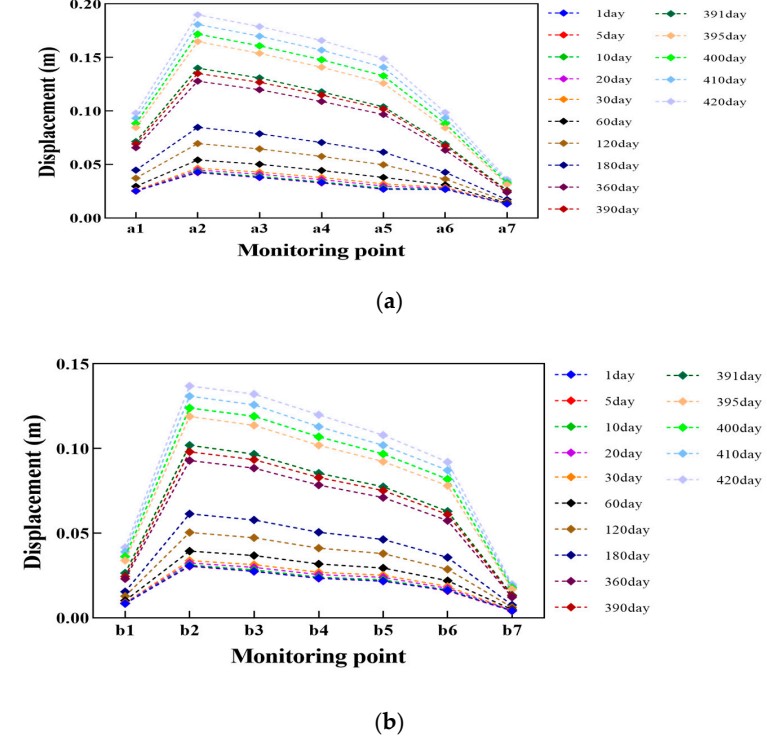

**Figure 16.** Deformation curves of the landslide slope surface and sliding zone. (**a**) Slope surface displacement, (**b**) Sliding zone displacement.

Monitoring points were positioned in downward intervals of 5 m in the rear (point a2), middle (point a4), and front (point a6) regions of the slope (Figure 17). Figure 17 shows that the sliding zone displacement was less than the sliding mass displacement; that is, the bedrock showed no significant displacement changes. The sliding zone deformation rate was also less than that of the sliding mass. At the back of the landslide, the maximum deformation occurred 5 m above the sliding zone. In the middle of the landslide, the deformation within the sliding mass was essentially the same and extended over the sliding area. At the front edge of the landslide, the deformation rate of the sliding mass was slightly less than that of the sliding zone, indicating that the location of the slope foot played a role in blocking the sliding.

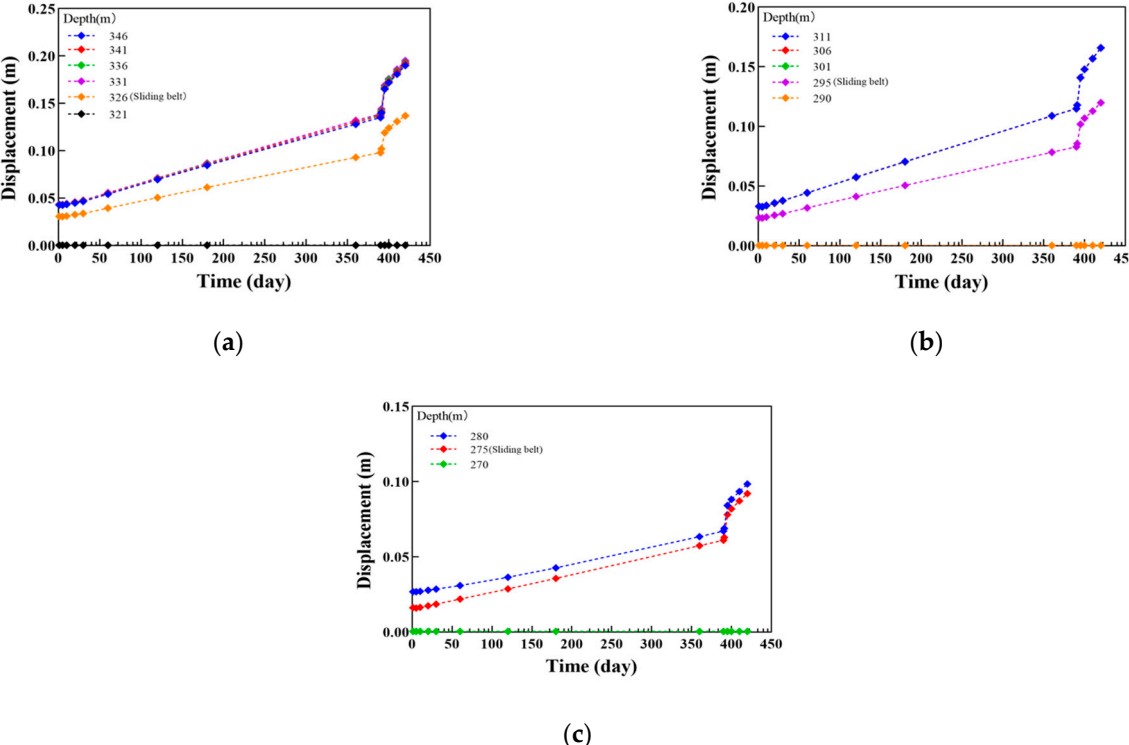

**Figure 17.** Displacement monitoring curves at different locations and depths in the slope. (**a**) Rear region of the slope, (**b**) Middle region of the slope, (**c**) Front region of the slope.

## 6. Discussion and Conclusions

Previous studies typically performed conventional triaxial creep tests on rock and soil using small specimen sizes (e.g., 5 cm in diameter; 10 cm in height); however, little research has been conducted on creep in sliding zones containing large particle sizes. In this study, natural and saturated triaxial creep tests were performed to investigate the creep behavior of the Huaipa landslide sliding zone, which has a high rock content. A creep study was conducted using a large triaxial creep apparatus with a specimen size of 30 cm in diameter and 60 cm in height, allowing for a maximum particle diameter of 6 cm. This ensured that the specimens were similar to the natural gradation state, thus making the creep deformation test specimens a more realistic representation of those in the natural state.

FLAC3D was used to study the disaster evolution mechanism of the Huaipa landslide. The results show that, when considering only the instantaneous strength and deformation, the slope only experiences unloading uplift at its foot and no overall sliding occurs. After considering the time effect, the slope creeps and slides, with tension damage appearing at the trailing edge, which is consistent with the deformation law. Tension fissures formed on the slope surface within one year of excavation and large-scale slides occurred after one

month of rainfall. A large number of natural slopes are stable in the short term but unstable after long-term creep deformation. Examples include the Jiudianxia reservoir Yanziping landslide [43], the Huangniba landslide [44], and the Santa Trada landslide, Bozzano [45]. Therefore, the creep characteristics of the rock and soil materials should be fully considered to accurately analyze the deformation and failure characteristics of landslides and evaluate slope stability.

In nature, the strength parameter of rock mass is expected to gradually decay with time because of its rheological properties [46]. The results presented here, based on particle fragmentation analysis during sliding zone creep processes, show that particle breakage is an important reason for the weakening of the rock and soil strength with time. Further research concerning the correlation between particle fragmentation, creep time, and strength weakening is needed in subsequent studies.

The long-term strength and residual strength of rock and soil are important mechanical indicators to assess whether ancient landslides will become reactive. Interestingly, the weakening mechanisms of both of these indicators have similarities. Fine crack damage within rock mass, fragmentation of the grain angles of soil–rock mixtures, and changes in the grain positions are important mechanisms for the weakening of rock and soil strength. The difference is that the residual strength is the result of strength weakening under a high-stress state, while the long-term strength is the result of strength weakening under a low-stress state. The long-term strength is therefore higher than the residual strength. The stress state of the sliding zone must be fully considered in slope stability evaluations and a reasonable strength index must be selected. Research on the strength loss of rock and soil under high- to low-stress states is also key to revealing the mechanism of landslide initiation damage and landslide reactivation. Technological developments have allowed an increasing number of studies to address changes in the particle position and fragmentation process of soil–rock mixtures [47], thereby gradually clarifying their strength weakening mechanisms.

Large-scale triaxial creep tests and three-dimensional numerical simulation analysis were conducted here to study (1) the creep characteristics of the Huaipa landslide sliding zone, which consisted of strongly weathered argillaceous sandstone, and (2) the mechanism of the disaster evolution of the slope under excavation and rainfall. The following conclusions were obtained:

(1) The strongly weathered argillaceous sandstone clearly exhibits creep properties, and the instantaneous strain and creep strain of the specimens can be observed in the tests. With increasing stress levels, the damage process of the specimens follows decay creep → steady creep → accelerated creep. The creep process of the strongly weathered argillaceous sandstone essentially involves the slow adjustment of internal particles and release of stress under the action of an external force, accompanied by complex strength loss and recovery. Via particle sliding, rotation, and fragmentation, the final state of particle adjustment minimizes the contact stress of each particle under a certain stress condition. There is no positional movement or particle fragmentation, and the rheological deformation tends to stabilize.

(2) The stress level plays a decisive role in the creep characteristics of the strongly weathered argillaceous sandstone. Higher stress levels are associated with higher instantaneous displacement and creep displacement and longer times required to reach stable deformation. The stress level magnitude also determines the creep stage (decay creep, steady creep, or accelerated creep). When the stress level is low, the adjustment of the internal structure of the specimen is dominated by void compression and particle movement, whereas particle fragmentation occurs mostly at high stress levels when the particle contacts inside the specimen increase and the contact stress is high.

(3) Water content also has an important influence on the creep characteristics of the strongly weathered argillaceous sandstone. The creep specimen with 13% moisture content had a high frictional resistance to particle movement. The specimen deforms slowly and shows better stability than saturated specimens. In contrast, internal moisture in

the saturated creep specimen had a lubricating effect on particle movement, the particle adjustment process was fast, and the displacement increased sharply with increasing stress levels, thereby resulting in poor stability.

(4) The long-term strength of the strongly weathered argillaceous sandstone was approximately 62–66% of its instantaneous strength. The internal friction angle decreased by approximately 8° during deformation relative to that of the instantaneous strength, and the cohesion decreased by approximately 30%. The creep effect has an important influence on the strength of the strongly weathered argillaceous sandstone. Therefore, long-term rock strength must be fully considered in the stability calculations of all related engineering projects.

(5) Excavation of the bauxite mine at the slope front edge destroyed the support at the slope foot of the Huaipa landslide, leading to unloading and large uplift deformation at the slope foot. However, the slope did not experience significant overall sliding at that time, and thereafter underwent creep deformation. Continuous rainfall later saturated the sliding zone, at which point the slope deformation notably accelerated and the slope slid as a whole, leading to the landslide disaster. The simulation results show that the landslide can be divided into a front edge bulging zone, a central sliding zone, and a trailing edge tension zone. These findings are fully consistent with the analysis results of the field geological investigation, thus providing important insight on the creep deformation evolution process and landslide disaster mechanism under the actions of front edge excavation and rainfall.

**Author Contributions:** J.D. conducted the experiment, processed the data, and revised the article. Y.Z. completed the experiment, processed the data, and wrote the article. H.L. conducted the conception of the experiment and revised the article. J.Z. helped complete the experiment and revised the paper. Z.Z. helped complete the experiment and revised the paper. Q.C. helped complete the experiment and completed some of the numerical simulations. J.Y. completed some of the numerical simulations and revised the paper. All authors have read and agreed to the published version of the manuscript.

**Funding:** This research was funded by the National Key Research and Development Project of China (Grant Number 2019YFC1509704).

**Institutional Review Board Statement:** Not applicable.

**Informed Consent Statement:** Not applicable.

**Data Availability Statement:** All the data used in this research are easily accessible by downloading the various documents appropriately cited in the paper.

**Acknowledgments:** This research work was sponsored by the National Key Research and Development Project of China (Grant No. 2019YFC1509704), the National Natural Science Foundation of China (Grant Nos. U1704243, 41741019, 41977249, and 42090052), and the Henan Province Science and Technology research project (Grant No. 192102310006).

**Conflicts of Interest:** Authors have no conflict of interest to declare.

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
