# Peer review of "Creep Characteristics of a Strongly Weathered Argillaceous Sandstone Sliding Zone and the Disaster Evolution Mechanism of the Huaipa Landslide, China"

_applsci, doi:10.3390/app13158579_

Round 1
Reviewer 1 Report
Review: Creep characteristics of a strongly weathered argillaceous sandstone sliding zone and the disaster evolution mechanism of the Huaipa landslide, China
This research focuses on creep characteristics of a strongly weathered argillaceous sandstone sliding zone. I believe that this paper will be of interest to the international readers if authors clarify more and improve the text. I recommend accepting and publishing this manuscript after minor revision. Below are the necessary corrections to improve the article:
1- References must be updated.
2- some sentences do not have any references that must be added.
3- Numbers in Tables 3 and 4 are illogical. What is the phrase “E”?
Minor editing of English language required
Author Response
Thank you very much for your valuable suggestions, which have improved the expression of the article. All comments have been corrected in the text. The corrected part has been marked in red.Please see the attachment.
Point 1: References must be updated.
Response 1: The references have been updated.
Point 2:some sentences do not have any references that must be added.
Response 2: All references have been added.
Point 3:Numbers in Tables 3 and 4 are illogical. What is the phrase “E”?
Response 3: Table 3 has been illustrated using a grain size distribution curve.

Reviewer 2 Report
The manuscript presented an experimental and numerical study of the creep behaviors of strongly weathered argillaceous sandstone. The authors may consider the following comments for further improvements.
Table 3, the authors may consider to present the PSD using a curve for better visualization, instead of using a table. Same comment applies to Table 5 & 6.
Fig. 1, it is recommended to label the direction of the ground movement in (a)-(c), also use blowouts to show the locations of the (b) and (c) in (a).
Fig. 5 was not mentioned in the manuscript.
What does SL# (# = 2, 4, 6, and 8) stands for in the legend of Fig. 7.
L213, how the three stages are distinguished from each other? Can the three stages be labeled in a figure to help the readers better understand the definitions and significance?
Fig. 7, it is better to scale the y-axis to the same range to help the readers better visualize the amount of deformations in all the tests.
Section 3.2.2, it is better to plot the strain as a function of each of the influence factors. It is hard for the reader to see from Fig. 7 the percentage of instantaneous strain to the total strain, for example.
What does the SL# (# = 1, 2, 3, and 4) mean in the legend of Fig. 9?
In Fig. 9, is the time representing the simulation time or the time simulated? For example, at the maximum time of 10000 min, does the simulation take that long to finish or does it take much longer (or shorter) to simulate that time?
Table 9 is hard to read. Why there are three rows for "sliding zones" with different material properties?
Language is fine. Minor edits are recommended.
Author Response
Thank you very much for your valuable suggestions, which have improved the expression of the article. All comments have been corrected in the manuscript. The corrected part has been marked in red.Please see the attachment.
Point 1: Table 3, the authors may consider to present the PSD using a curve for better visualization, instead of using a table. Same comment applies to Table 5 & 6
Response 1:Table 3、5 and 6 have been illustrated using a grain size distribution curve.
Point 2: Fig. 1, it is recommended to label the direction of the ground movement in (a)-(c), also use blowouts to show the locations of the (b) and (c) in (a).
Response 2:Changes and annotations have been made in the figure.
Point 3: Fig. 5 was not mentioned in the manuscript.
Response 3:Figure 5 has already been mentioned in the article.
Point 4: What does SL# (# = 2, 4, 6, and 8) stands for in the legend of Fig. 7.
Response 4:I'm very sorry, but due to my unclear expression, you are confused.Thank you for your careful reminder.
In this paper,six sets of tests were performed on specimens with a diameter of 30 cm and height of 60 cm. Conventional triaxial tests were conducted to determine the ultimate deviator stress (qf) at the time of damage. Graded loading was applied to reduce the creep test time and dispersion of specimens. The selected creep stress levels were 0.2qf, 0.4qf, 0.6qf, 0.8qf, and 1.0qf. In the following, we abbreviate the stress level as 'SL' and the corresponding unused stress level is abbreviated as ‘SL=2’、 ‘SL=4’ 、‘SL=6’ and ‘SL=8’.
Point 5: L213, how the three stages are distinguished from each other? Can the three stages be labeled in a figure to help the readers better understand the definitions and significance?
Response 5:In order to clarify this issue more clearly, I have added some references and provided explanations.
Research suggests that when the creep stress is less than the long-term strength, the soil sample exhibits a decaying creep mode; When the creep stress is greater than the long-term strength, the soil sample exhibits a creep or non decaying creep mode as the strain rate increases .
The curve diagram of the full creep process has been added
Point 6: Fig. 7, it is better to scale the y-axis to the same range to help the readers better visualize the amount of deformations in all the tests.
Response 6:Adjustments have been made to Figure 7
Point 7: Section 3.2.2, it is better to plot the strain as a function of each of the influence factors. It is hard for the reader to see from Fig. 7 the percentage of instantaneous strain to the total strain, for example.
Response 7:Thank you for your valuable suggestion.I have added tables for Instantaneous strain and total strain
Point 8: What does the SL# (# = 1, 2, 3, and 4) mean in the legend of Fig. 9?
Response 8:Sorry for any confusion caused by my negligence. SL is consistent with Figure 7 and has been renumbered
Point 9: In Fig. 9, is the time representing the simulation time or the time simulated? For example, at the maximum time of 10000 min, does the simulation take that long to finish or does it take much longer (or shorter) to simulate that time?
Response 9:Figure 9 shows the actual creep duration, not the calculation time. In the simulation process, the calculation time can be represented as a longer creep time, so there is no need for a longer time during calculation. The calculation time for calibration of small samples is about 30 minutes. When calculating the 3D model, maintain the same correspondence between calculation time and creep time for analysis. Due to the longer creep time during three-dimensional analysis, the corresponding calculation time is also longer. It taked approximately 24 hours to complete the calculation
Point 10: Table 9 is hard to read. Why there are three rows for "sliding zones" with different material properties?
Response 10:Because in the simulation process, the instantaneous excavation process of the landslide, unsaturated creep process, and saturated creep process after rainfall were considered. In the analysis of three working conditions, the mechanical parameters and constitutive model of the sliding zone show significant changes, so there are three types of material parameters in this area.

Round 2
Reviewer 2 Report
The authors have addressed my previous comments and questions. I have not further comments.
Minor edits are needed.